# ATR-FTIR Spectroscopy Combined with Multivariate Analysis Successfully Discriminates Raw Doughs and Baked 3D-Printed Snacks Enriched with Edible Insect Powder

**DOI:** 10.3390/foods10081806

**Published:** 2021-08-05

**Authors:** Nerea García-Gutiérrez, Jorge Mellado-Carretero, Christophe Bengoa, Ana Salvador, Teresa Sanz, Junjing Wang, Montse Ferrando, Carme Güell, Sílvia de Lamo-Castellví

**Affiliations:** 1Departament d’Enginyeria Química (DEQ), Campus Sescelades, Universitat Rovira i Virgili, Av. Països Catalans, 26, 43007 Tarragona, Spain; nerea.garcia@urv.cat (N.G.-G.); jorge.mellado@urv.cat (J.M.-C.); christophe.bengoa@urv.cat (C.B.); junjing.wang@urv.cat (J.W.); montse.ferrando@urv.cat (M.F.); carme.guell@urv.cat (C.G.); 2Instituto de Agroquímica y Tecnología de Alimentos (IATA-CSIC), C/Catedràtic Agustín Escardino Benlloch, 7, 46980 Paterna, Spain; asalvador@iata.csic.es (A.S.); tesanz@iata.csic.es (T.S.)

**Keywords:** insect powder, authentication, 3D food printer, mid-infrared spectroscopy, multivariate analysis

## Abstract

In a preliminary study, commercial insect powders were successfully identified using infrared spectroscopy combined with multivariate analysis. Nonetheless, it is necessary to check if this technology is capable of discriminating, predicting, and quantifying insect species once they are used as an ingredient in food products. The objective of this research was to study the potential of using attenuated total reflection Fourier transform mid-infrared spectroscopy (ATR-FTMIR) combined with multivariate analysis to discriminate doughs and 3D-printed baked snacks, enriched with *Alphitobius diaperinus* and *Locusta migratoria* powders. Several doughs were made with a variable amount of insect powder (0–13.9%) replacing the same amount of chickpea flour (46–32%). The spectral data were analyzed using soft independent modeling of class analogy (SIMCA) and partial least squares regression (PLSR) algorithms. SIMCA models successfully discriminated the insect species used to prepare the doughs and snacks. Discrimination was mainly associated with lipids, proteins, and chitin. PLSR models predicted the percentage of insect powder added to the dough and the snacks, with determination coefficients of 0.972, 0.979, and 0.994 and a standard error of prediction of 1.24, 1.08, and 1.90%, respectively. ATR-FTMIR combined with multivariate analysis has a high potential as a new tool in insect product authentication.

## 1. Introduction

The world population is increasing dramatically mainly as a result of the quality-of-life improvement in developing countries and will reach over 9.8 billion by 2050 [1]. An increasing demand for protein-rich sources will be a threat to world food and feed availability [2]. Nowadays, animal protein sources come mostly from livestock, such as poultry, swine, and cattle. Stockbreeding requires large spaces and large quantities of natural resources and also produces significant greenhouse emissions among other contaminants [3].

Insects have been proposed as a suitable alternative to conventional livestock [4]. In fact, insect breeding needs fewer resources compared to conventional livestock: water consumption is lower; less space occupancy is required; and insects can be fed using food waste products, such as potato peels, rotten fruits, and bakery by-products [5].

Moreover, when comparing the protein content of both insects and livestock, the first one shows similar or even higher amounts of protein. For instance, the amount of protein in 100 g of fried grasshoppers is almost triple compared to the protein content in 100 g of grilled beef [6].

Besides the high protein content, most insects have high amounts of polyunsaturated fatty acids, vitamins, and micronutrients (e.g., calcium, iron, and phosphorus) essential for human life [7]. However, it has been observed that the consumption of insects can be risky due to the presence of possible pathogens, antinutrients, and allergenic substances, assessed in several cross-reactivity studies [8,9].

Before being able to open the market to the commercialization of insects, the European Commission needed to confirm that their consumption was totally safe for human health. Unfortunately, in 2019, there was not enough information available about the potential risk associated with insect consumption [10]. Since then, several studies have been performed to evaluate the composition of common edible insect species. In early 2021, the European Food Safety Authority (EFSA) revealed its opinion, positioning itself in favor of the consumption of a very well-studied insect species, *Tenebrio molitor* [11]. A recent study in the European insect market, conducted by the International Platform of Insects for Food and Feed (IPIFF), forecasts that the following insect species to be approved will be the lesser mealworm (*Alphitobius diaperinus*) and the migratory locust (*Locusta migratoria*) among others [12]. The final approval by the European Commission of the use of *T. molitor* as a new ingredient in food products is smoothing the path to the introduction of insects in Western diets [13].

Another challenge to overcome is insect consumption in Western countries. One strategy that has been followed to introduce insects in Western diets is the use of edible insect powders (i.e., dehydrated insects that have been ground to obtain a fine powder) as ingredients for innovative food product design. However, the use of insect powders as ingredients can lead to adulteration and fraud.

There are several techniques used by the food industry to verify the authenticity of their products. Molecular techniques (e.g., genomics, proteomics, and polymerase chain reaction with denaturing gradient gel electrophoresis (PCR-DGGE)) have been widely used for origin authentication. Despite their high sensitivity, further characterization of markers is required for real unknown samples [14]. In the case of complex food matrices, several fingerprinting techniques, such as chromatographic techniques (e.g., high-performance liquid chromatography (HPLC) and mass spectrometry (MS)), have been selected since they are rapid and easy to execute [15]. As an example, matrix-assisted laser desorption/ionization coupled with a time-of-flight mass spectrometer (MALDI-TOF) has been used as a method for the authentication and classification of several commercial edible insect powders [16]. Although MALDI-TOF mass spectrometry has been proven to be a useful technique for insect powder authentication, it has several drawbacks, such as time of analysis and cost, that could hinder its implementation in the food industry.

Other fingerprinting techniques applied in food authentication are based on rotational–vibrational spectroscopy (infrared and Raman spectroscopy). Fourier transform infrared (FTIR) spectroscopy has been applied in different food matrices, such as juices, meat, and extra virgin olive oil authentication, due to the ease of obtaining powerful results quickly [17,18]. Nevertheless, fingerprinting techniques produce huge volumes of data that need to be processed. For this reason, multivariate analysis is essential to process the data collected based on fingerprint techniques [19].

The objective of the present work was to study the potential of using attenuated total reflectance Fourier transform mid-infrared spectroscopy (ATR-FT-MIR) combined with multivariate analysis to rapidly discriminate and predict the concentration of *A. diaperinus* and *L. migratoria* powder added into a raw dough and 3D-printed baked snacks.

## 2. Materials and Methods

### 2.1. Materials

Lesser mealworm (*Alphitobius diaperinus*) and migratory locust (*Locusta migratoria*) powder from three different batches (100 g each) were supplied by Kreca Ento-Food BV (Ermelo, The Netherlands). The chickpea flour was purchased from La Finestra Sul Cielo S.A. (Madrid, Spain) and the hot madras curry powder from Westmill Foods (London, UK). The extra virgin olive oil was obtained from Hacienda Ortigosa S.L. (Navarra, Spain) and the salt from Sal Costa, S.L.U (Barcelona, Spain).

### 2.2. Dough Preparation

The ingredients selected to create a blank dough (B) were chickpea flour (46.3 wt%), water (39.4 wt%), extra virgin olive oil (11.6 wt%), curry powder (1.8 wt%), and salt (0.9 wt%). This dough formulation was made to obtain gluten- and dairy-free tasty dough with good printability. Based on this formulation, different quantities of *A. diaperinus* or *L. migratoria* powder were added, replacing 10% (A_d1_, L_d1_), 20% (A_d2_, L_d2_), or 30% (A_d3_, L_d3_) of the total amount of chickpea flour used (Table 1). Both insect powders were previously milled using an electric coffee grinder (TM-CG-03, Vilapur, Larkhall, UK) for 1 min. The milling process reduced the particle size of *A. diaperinus* powder from D_[4,3]_ values of 677 µm (span of 2.0) to 310 µm (span of 2.2). For *L. migratoria* commercial powder, almost 30% of the particles were bigger than 2000 µm, and, after the milling process, a D_[4,3]_ of 967 µm (span of 1.1) was obtained (Appendix A). A total of 100 g was prepared for each formulation, mixing all the ingredients with a hand blender (HB-10C.019A, HAEGER SPAIN, Barcelona, Spain) for 2 min.

### 2.3. 3D Printing and Post-Processing Analysis

A portable 3D food printer (Focus, ByFlow Company, Eindhoven, The Netherlands) was used to print the blank dough (B) and several doughs enriched with 4.6% (A_d1_), 9.3% (A_d2_), and 13.9% (A_d3_) of *A. diaperinus* powder. In the case of the dough enriched with *L. migratoria*, the printing process could not be carried out because the average particle size obtained in the ground *L. migratoria* powder was larger than the maximum size of the nozzle of the 3D printer used. The printing was performed at room temperature using a Voronoi circle model of 100 × 100 × 10 mm (Figure 1a). A total of 32.0 g of each dough was printed using a 1.6 mm nozzle at a speed of 30 mms^−1^. Then, the dough was baked at 180 °C for 12 min using the ventilation mode of a vapor oven (3HV469X/02 model, Balay, Pamplona, Spain). Each formulation was made in triplicate.

In order to find out the amount of water lost during the baking process, the total weight of the 3D-printed snack was measured before and after baking it (Table 2) (Figure 1b,c). The concentration of the insect powder was recalculated based on the percentage of water loss of the 3D-printed baked snacks.

### 2.4. Spectral Data Acquisition by FTIR

Spectral profiles were collected in the mid-infrared (MIR) region (4000–800 cm^−1^) with 8 cm^−1^ resolution using portable spectrometer Cary 630 (Agilent Technologies Spain SL, Madrid, Spain) equipped with a single bounce ATR diamond crystal accessory and a deuterated triglycine sulfate (DTGS) detector. Spectral information acquisition was carried out using MicroLab PC software (Agilent Technologies SL, Madrid, Spain). The final spectra were obtained from the average of 128 scans to improve the signal-to-noise ratio, and a background scan was taken before every sample scan to avoid noise in the spectral data from the environment.

For the dough spectral acquisition, 4 mg of dough was placed on the sample stage using a rolling pin (Excel Blades Corp, Paterson NJ, USA). Then, the sample was dried by vacuum to remove the water contribution in the final spectra.

The baked snacks were milled using an electric grinder (TM-CG-03, Vilapur, Larkhall, UK). Then, 4 mg of ground snack was placed in the sample stage. To enhance the contact between the diamond crystal and the sample, a press clamp was used, standardizing the layers’ width of the sample over the detector.

Spectral data from three different batches (prepared at three different days) of each blank dough, dough enriched with 4.6, 9.3, and 13.9% of *A. diaperinus* and *L. migratoria,* and 3D-printed baked snack with 7.2, 14.4, and 21.9% of *A. diaperinus* were obtained, collecting 10 spectra per day and per sample. A total number of 30 spectra were obtained for each type of sample at room temperature.

### 2.5. Multivariate Analysis

Multivariate analysis and data preprocessing were performed using a chemometric software (Pirouette, version 4.5. Infometrix Inc., Bothell, WA, USA). Spectral data from the doughs were mean centered, vector length normalized and transformed using second derivative polynomial-fit Savitzky–Golay function (13 points). The spectral data from the ground snacks were mean centered, transformed using multiplicative scatter correction (MSC) and second derivative polynomial-fit Savitzky–Golay function (13 points). A statistical Principal Component Analysis (PCA)-based supervised algorithm, soft independent modelling of class analogy (SIMCA), was used to create a discrimination and classification model with the spectral data obtained [20]. In order to detect potential outliers, sample residuals and the Mahalanobis distances were taken into account [21]. Three different outputs were used to interpret SIMCA models created: class projections (i.e., 3-dimensional PCA score plots), interclass distances, and discriminating power. Total misclassifications were analyzed and interpreted for the input data. Models were validated using a predicted set of samples, excluding 5 spectra per sample from the initial input data and creating new models with the remaining data.

The pre-treated spectra were also analyzed by partial least squares regression (PLSR) that was cross-validated (leave-one-out, internal validation approach) to generate calibration models. The same transformations used for the creation of the SIMCA models were applied to the spectral data used for the PLSR analysis: mean centered, vector length normalized, and transformed by second derivative polynomial-fit Savitzky–Golay function (13 points) in the case of the dough and MSC and second derivative polynomial-fit Savitzky–Golay function (13 points) for the 3D-printed baked snacks. Thus, the x variable was the absorbance per each wavenumber, and the y variable (reference data) was the percentage of edible insect powder added to the dough, or, in the case of the 3D-printed snacks, the percentage of insect powder present in the final product. Furthermore, a predicted set of samples was used in order to validate the model, temporarily leaving out 5 spectra per sample from the training set. All the models were evaluated in terms of regression vector, standard error of cross-validation (SECV), standard error of calibration (SEC), determination coefficient (R^2^), and outlier diagnostics [22].

## 3. Results and Discussion

### 3.1. Spectral Information of Ingredients and Doughs

Previous studies have shown the ability of MIR spectroscopy to provide information about the chemical composition of complex samples, such as food matrices, correlating different IR bands with specific functional groups [23]. Moreover, it is important to analyze all the ingredients alone to help elucidate the origin of the IR bands associated with certain components in these mixtures (Figure 2).

Comparing the spectral information from the ingredients used in the dough and snack formulation (Figure 2), it can be observed that chickpea, curry, *A. diaperinus,* and *L. migratoria* powder showed a broad IR band from 3000 to 3500 cm^−1^, which originated from the stretching of O-H bonds most likely coming from carbohydrates, fiber, and water [24]. Moreover, all these spectra exhibited two narrower IR bands at around 2900 and 2850 cm^−1^, characteristics of asymmetrical and symmetrical stretching vibrations from C-H bonds from methyl groups presumably caused by lipids or, in the case of insect powders, lipids and chitin. Another common IR band between all the ingredients was observed around 1740 cm^−1^. This IR band can be associated with the stretching of C=O bonds from ester groups related to lipids [25,26]. Chickpea flour and insect powders presented IR bands in the region of 1650 and 1500 cm^−1^, and these were related to C-N stretching, C=O vibrations of N-acetyl groups, and N-H bending from amide II groups, most likely from the presence of proteins. In the case of insect powders, these IR bands can also be associated with chitin presence [27]. Finally, a distinctive IR band at 1100–900 cm^−1^ from C-O stretching vibrations, most probably corresponding to the presence of non-structural carbohydrates, such as starch and sugars, was also detected in the chickpea and curry powder spectra [28].

The raw MIR spectra of the blank dough (B) and the doughs with different concentrations of *A. diaperinus* (A_d1_, A_d2_, and A_d3_) and *L. migratoria* (L_d1_, L_d2_, and L_d3_) with their respective transformed spectra are shown in Figure 3. Differences between the dough samples are not easy to detect from the raw spectra, since all of them exhibit the same IR bands related to the presence of lipid and chitin (2900 and 2850 cm^−1^), lipid (around 1700–1740 cm^−1^), protein (1650 and 1500 cm^−1^), and carbohydrates and chitin (from 1200 to 900 cm^−1^), with a similar absorbance.

All this information was important to decide the spectral region selected for the building up of the SIMCA models.

First of all, the region from 4000 to 3000 cm^−1^ was excluded from the model due to the high moisture content of the dough samples, trying to avoid discriminations based on water content. Furthermore, the IR bands between 2700 and 1800 cm^−1^ were also omitted to reduce the noise impact in the spectral analysis caused by the crystal.

Moreover, for all the formulations, the amount of insect powder added was increasing at the same time that the proportion of chickpea flour was decreasing due to the substitution of ingredients shown in Table 1. As can be seen in the raw spectra of the chickpea flour in Figure 2, the signal in the carbohydrate region is quite intense compared to the other ingredients. Therefore, since the aim of our research was to determine if it was possible to discriminate the presence of insect powder in the designed product and in view of the fact that the region from 1200 to 800 cm^−1^ was strongly influenced by the variable quantities of chickpea flour, it was considered more appropriate to exclude this region from the model. Finally, the spectral region chose for the data analysis was the one between 3000 to 2700 cm^−1^ and 1800 to 1200 cm^−1^.

### 3.2. Discrimination and Classification of Doughs by ATR-FT-MIR Combined with SIMCA

Initially, two different four-class SIMCA models were built up to obtain classification models to discriminate the blank and the doughs enriched with different amounts of *A. diaperinus* (A_d1_, A_d2,_ and A_d3_) or *L. migratoria* (L_d1_, L_d2,_ and L_d3_) powders and obtain information about their biochemical differences.

To obtain these models, the original variables are replaced by linear combinations of the same variables called factors, helping to reduce the dimensionality of the data without losing information [29]. The number of factors was chosen to achieve a minimum of 90% of the variance in each class of both models (Table 3).

SIMCA class projection is a three-dimensional scatter plot that gives information about the similarity of the spectral data (Figure 4). Each point in the class projection represents a spectrum that belongs to a cluster (represented with an ellipse) with 95% of confidence. Those groups that seem closer will be more similar to each other unlike those that will appear further away, which will be much different [30].

The transformed spectra (3000–2700 cm^−1^ and 1800 to 1200 cm^−1^ region) of doughs enriched with *A. diaperinus* powder (Figure 4a) or *L. migratoria* (Figure 4b) showed non-overlapping and well-separated clusters allowing accurate dough classification for each type. SIMCA’s misclassification algorithm for doughs enriched with *A. diaperinus* or *L. migratoria* powders showed zero misclassifications, indicating that the training set for each model was homogeneous, and all samples were correctly classified into their assigned classes.

Discrimination power plots (Figure 5) show which are the major IR bands that contributed to the development of the classification models. Despite the fact both models were made with distinct insect species, similar IR bands were responsible for the differentiation of the clusters. As can be seen in Figure 5, the IR bands mainly responsible for the discrimination of the models were related to the stretching of C-H bonds from methyl groups associated with lipid content (2959 cm^−1^ and 2877 cm^−1^), the C=O stretching of esters of lipids (1777 cm^−1^), amide I groups most likely from proteins and chitin (1647 cm^−1^ till 1546 cm^−1^), and bending vibrations of the CH_2_ and CH_3_ aliphatic groups (1427 cm^−1^) that can be related to polysaccharides and sugars [31,32].

Interclass distance (ICD) is a ratio of Euclidean distances between two classes that indicate the similarities and/or dissimilarities between them [33]. ICD values above 3 are considered significant to discriminate two clusters of samples as distinct classes [34]. As shown in Table 3, ICD values between the clusters of dough enriched with *A. diaperinus* powder increased with the amount of insect powder added, ranging from 3.1 to 12.2. In the case of dough enriched with *L. migratoria*, the same tendency is observed, showing increasing ICD values with the amount of insect powder added from 5.8 to 13.6 (Table 4). The fact that ICD values increased with the amount of insect powder added shows that this ingredient is responsible for the dough classification [35].

Model validation using an independent set of spectra (five spectra not included in the SIMCA model from each class) showed a 100% correct classification for each type of dough in both models tested. The specificity of each model was also evaluated by making predictions of the *L. migratoria* spectra of each class into the four-class SIMCA *A. diaperinus* dough model and by using this one to classify the spectra (Table 5). The same procedure was used to evaluate the *L. migratoria* model. These results confirmed the ability of the models developed to discriminate dough samples depending on the species of insect used (*A. diaperinus* and *L. migratoria*) and the quantity of insect powder added [36].

It was also important to study if it was possible to differentiate between the type of insects used to make the doughs. For this purpose, a six-class SIMCA model was created using the IR data of doughs with different concentrations of *A. diaperinus* and *L. migratoria* powders. The number of factors chosen for this model as well as their cumulative variance is shown in Table 6.

The class projection plot (Figure 6) showed distinctive clustering patterns and six well-defined classes, which are closer to those with a low amount of insect powder and far away from the clusters with the highest quantity of insect powder.

The discriminating power plot (Figure 7) showed that, as in the previous models, the main discrimination was related to lipids (C=O stretching of esters from lipid content at 2877 cm^−1^ and 1722 cm^−1^ to 1710 cm^−1^), protein or/and chitin content (amide I from proteins and/or chitin at 1606 cm^−1^ and 1535 cm^−1^), and polysaccharides (1461 cm^−1^). This reinforces the idea that these components might be important factors in the discrimination of different insect species.

Moreover, the maximum discriminating power obtained in the *A. diaperinus* and *L. migratoria* dough models was almost triple that obtained in the *A. diaperinus* vs. *L. migratoria* dough model (Figure 7). This information reveals that both types of dough display remarkably similar composition despite being made with different insect powders.

The ICD values observed in the model that compare the different insect doughs exhibit the same trend as the individual models (see Table 4). The ICD value increases with the insect powder content in the dough regardless of the type of insect powder used (Table 7).

The validation of the model was carried out using a new set of spectra, five for each class. A 100% correct classification was obtained from all the spectra in all the tested classes. Therefore, the capability of this model to discriminate between *A. diaperinus* and *L. migratoria* powders in a food matrix was corroborated.

### 3.3. Spectral Information of 3D-Printed Snacks

Another objective proposed in this research was to study if the insect powder could be discriminated in the final product after being extruded by a 3D printer and going through a baking process. Figure 8 shows the raw spectra of 3D-printed baked snacks made with different amounts of *A. diaperinus* powder.

Thermal treatments, such as baking, are commonly used processes in the food industry and in food preparation [37]. Some of these methods can help in elongating the shelf-life of the processed products and improve the digestibility and bioavailability of proteins. Nevertheless, thermal treatments can negatively affect a wide diversity of molecules (e.g., lipid oxidation, protein denaturation, and vitamin solubilization) [38,39].

When comparing the raw spectra of these snacks with the raw spectra of the corresponding doughs in Figure 2, we can observe that the broad IR band from 3000 to 3500 cm^−1^ was reduced. This IR band range is linked to the stretching of O-H bonds, and its decreasing signal can be related to the water loss reported during the baking process [40].

The oxidation of unsaturated lipids is an autocatalytic reaction enhanced by thermal processes [41]. However, the signal on the IR bands around 2900, 2850, and 1740 cm^−1^, most likely related to the presence of lipid, seems to remain similar to the one shown in the dough spectra.

Moreover, another spectrum region that showed a distinct signal was the region at 1650 and 1500 cm^−1^, presumably from protein or chitin content. A plausible explanation for this fact could be the denaturation of proteins due to the heating process at high temperatures [42]. Studies on protein denaturation in milk-derived products reveal that certain regions of the spectrum related to protein content (1700–1695 cm^−1^ (aggregated β-sheets), 1645 cm^−1^ (random structure), and 1609 cm^−1^ (side chains)) have been affected by heat treatments, altering the spatial conformation of the proteins and, as a consequence, the intensity of the related IR bands [43]. Another approach for this variation on the IR bands could be the presence of Maillard reaction products. Maillard reaction is a non-enzymatic reaction, binding amino components and reducing sugars with covalent bonds, obtaining aromatic compounds as a result [44].

Finally, a characteristic IR band at 1100–900 cm^−1^, probably from carbohydrates, can be observed. As has been commented before, thermal treatments can lead to changes in different nutrients. In the case of carbohydrates, these can be involved in several complex reactions. One of these reactions is the hydrolyzation of long carbohydrates, such as starch, obtaining reduced sugars and molecules that can enhance other reactions, such as the above-mentioned Maillard reaction. Regarding the starch content, this macromolecule can also undergo a process called gelatinization, changing its structure from ordered to disordered, affecting its solubilization. Furthermore, free sugars can caramelize due to the dehydration caused by the thermal treatment, contributing to the browning process [42,45].

All these changes produced by the aforementioned chemical reactions were reflected in the raw spectra of the snacks, which were used afterward to explain the origin of the IR regions responsible for the discrimination of the model created.

### 3.4. Discrimination and Classification of Snacks by ATR-FT-MIR Combined with SIMCA

A four-class SIMCA model was created to assess the differences between snacks made with an increasing amount of *A. diaperinus* powder. A total of two factors were selected to create this model, achieving a cumulative variance higher than 90% in all the classes of the model (Table 8).

In the case of the class projection of the *A. diaperinus* snacks, the model showed well-defined clusters. A two-axis plot was used to represent this model since only two factors were needed to discriminate the different classes (Figure 9).

The discriminating power plot of the *A. diaperinus* snack model (Figure 10) showed major discrimination of the IR band at 1647 cm^−1^, from amide I groups most likely from protein and chitin presence, and the IR bands at 1707, 1699, and 1610 cm^−1^, probably linked to different structural configurations of proteins affected by the thermal treatment. This fact can also explain the decrease in the discriminating power from the IR band at 1505 cm^−1^.

All the classes exhibited a good classification, showing ICD values over 3. In addition, this model shows the same trend as the previous models shown before of *A. diaperinus* or *L. migratoria* dough, increasing the ICD value with the increase in insect powder concentration (Table 9).

Five spectra from each class, that were not previously included, were used to validate the model. Each class obtained a 100% correct classification of the new spectra showing the capability of the model created to discriminate different baked snacks based on the concentration of *A. diaperinus* powder used in the formulation.

### 3.5. PLSR of Insect Powder Concentration in Doughs and Snacks

Based on the information obtained from the SIMCA models, a partial least square regression (PLSR) analysis was applied to study the potential of predicting the concentration of insect powder added to enrich the doughs and the snacks. PLSR models were built up using the MIR region between 3000–2700 cm^−1^ and 1800–1200 cm^−1^ (Figure 11).

The optimal number of factors was chosen based on a cumulative variance of the factors higher than 90%, a low standard error of prediction (SEP), and the lowest jaggedness possible. The SEP estimates the expected error of predicting an unknown sample, and the jaggedness is a measurement that quantifies the relevance of the noise compared to the overall signal [46]. A low number of factors can lead to a poor explanation of the data, obtaining high SEP and jaggedness values. On the other hand, a very large number of factors end up overfitting the data, causing the algorithm to perform inaccurately with new data sets [47]. In this case, the SEP will keep reducing its value but the jaggedness value will increase revealing a high impact of the noise in the model. Following these criteria, two factors were chosen for the PLSR models of *A. diaperinus* doughs (A_d_ PLSR), *L. migratoria* doughs (L_d_ PLSR), and the *A. diaperinus* snacks model (A_S_ PLSR). It is also important to mention that all the PLSR models obtained a high R^2^, showing an excellent prediction accuracy (Table 10).

The regression vector was taken into account in order to find out which were the most remarkable wavenumbers used to develop the PLSR model. The regression vector is the weighted addition, depending on the variance, of each of the wavelengths comprised in the model. The variable that contributes significantly to the prediction of the sample has a higher coefficient compared to those that do not.

In the case of the PLSR dough models, the regression vector had a similar profile (Figure 12). Both models showed that the IR bands that contributed most to the prediction of the percentage of insect powder were 1647 cm^−1^ and 1610 to 1606 cm^−1^, linked to amide I groups probably related to protein and chitin presence. Moreover, the IR band at 2877 cm^−1^, associated with the stretching of C-H bonds from methyl groups, most likely from lipid presence, was also detected. All these regions from the spectra were exhibited previously in the discriminating power from the *A. diaperinus* and *L. migratoria* SIMCA dough models explained above.

Prediction models from *A. diaperinus* dough and *L. migratoria* dough exhibited similar IR bands at 2922 and 2855 cm^−1^, associated with the stretching of C-H bonds from methyl groups (most likely from lipid presence), and 1535 cm^−1^, presumably from protein and chitin presence (amide I groups).

The same effect was found in the regression vector of the 3D-printed baked snack (Figure 13). The regression vector obtained from the PLSR snack model showed similar IR bands to those exhibited in the discriminating power of the *A. diaperinus* SIMCA snack model in Section 3.4. In the case of the regression vector, the IR bands with the highest contribution in the prediction model were probably related to protein, in particular to different secondary structures, such as aggregated β-sheets (1700–1695 cm^−1^), random structures (1645 cm^−1^), and exposed side chains (1609 cm^−1^), as a result of the thermal treatment during the baking process.

To properly evaluate if the IR bands present in the regression vectors were linked to the insect powder or/and chickpea flour, it was necessary to compare them not only with the pretreated spectra of ingredients (Figure 2) but also with the doughs and snacks spectral information (Figure 3 and Figure 8). In the case of the IR bands related to lipids (1740 cm^−1^) and lipids and chitin (the region between 3000 and 2700 cm^−1^), a slight increase in the signal was detected in the doughs as the amount of insect powder increased (Figure 3). Comparing to the transform spectra of raw ingredients (Figure 12c), the amount of lipids present in the chickpea flour was lower than that in both insect powders. Thus, the differentiation by the lipid region in the PLSR model was mainly related to the insect powder added to the dough. However, in the case of the protein region of the pretreated spectra from the doughs (between 16,500 and 1500 cm^−1^), most of the IR bands showed increasing absorbances while others exhibited a decreasing signal (Figure 3a,c). The pretreated spectra of chickpea flour and *A. diaperinus* and *L. migratoria* powders also showed IR bands at this region (Figure 12c). It is also known that different secondary structures of proteins absorb at different wavenumbers [48]. Thus, the differences in the absorbance of the protein region were not only due to the variation of the chickpea flour/insect powder ratio but also due to the type of proteins added, the bonds that conform them, and the media (dough or snack).

In the literature, a collection of models created with the PLSR algorithm using MIR and NIR spectral data have been successfully used to predict the concentration or amount of ingredients and products used by the food industry [49,50]. A portable FT-MIR spectrometer combined with multivariate analysis can be used for routine controls for authentication of standardization of quality control of *A. diaperinus* and *L. migratoria* products both during the fabrication process or at the end of the production line.

## 4. Conclusions

ATR-FT-MIR combined with multivariate analysis can be used as a rapid technique to discriminate edible insect powders used as ingredients for doughs and 3D-printed snacks. Moreover, using PLSR analysis, calibration models can be built up to easily predict the concentration of insect powder present in doughs and snacks.

Further work needs to be carried out to determine the feasibility of this method for detecting insect powders in other food matrices, either raw or cooked.

## Figures and Tables

**Figure 1 foods-10-01806-f001:**
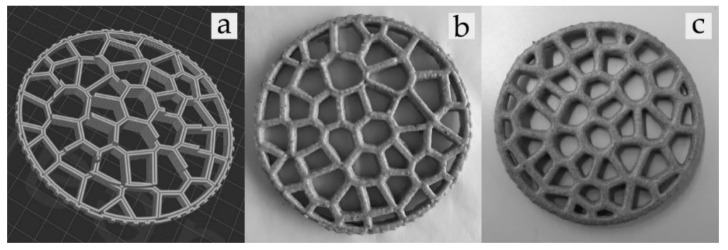
The following photos show (**a**) a 3D visualization created by Webgcode of the Voronoi circle model, (**b**) 3D-printed raw snack, and (**c**) 3D-printed baked snack.

**Figure 2 foods-10-01806-f002:**
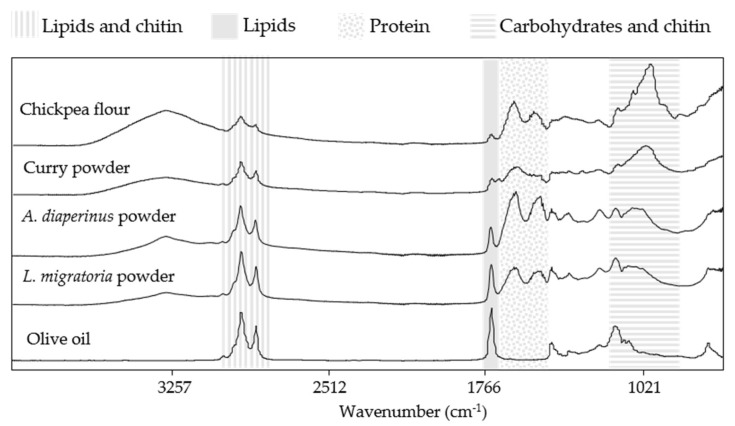
Raw mid-infrared spectra (4000 to 800 cm^−1^) from the ingredients used to prepare the doughs and the snacks.

**Figure 3 foods-10-01806-f003:**
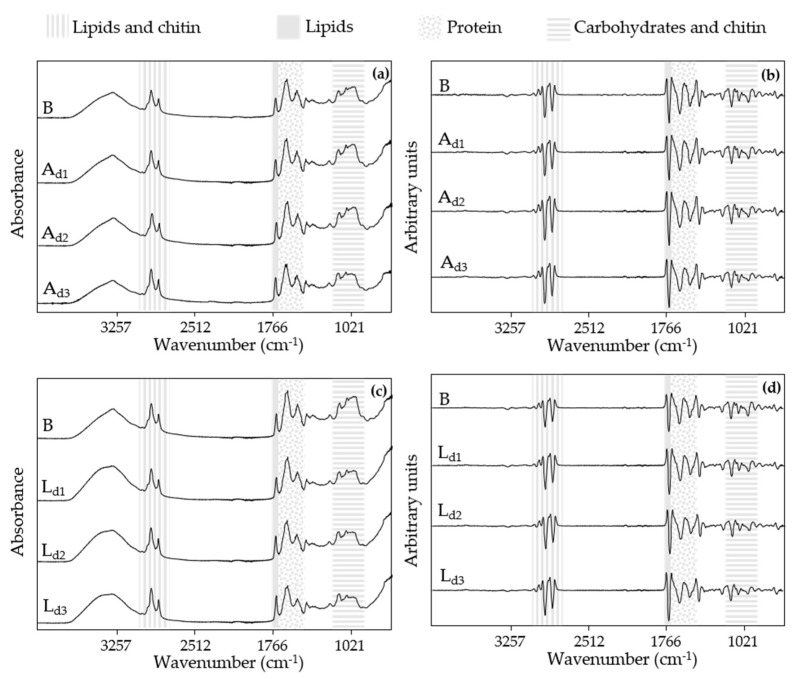
Raw MIR spectra (4000 to 800 cm^−1^) from doughs with (**a**) different concentrations of *A. diaperinus* (A_d1_, A_d2_, and A_d3_) and (**b**) the transformed spectra (vector length normalized and second derivative, 13 points). Graphs (**c**) show the raw mid-infrared spectra of different concentrations of *L. migratoria* (L_d1_, L_d2_, and L_d3_) and (**d**) the transformed spectra. Abbreviations used: B, blank dough with 0% insect powder; A_d1_, dough with 4.6% of *A. diaperinus* powder; A_d2_, dough with 9.3% of *A. diaperinus* powder; A_d3_, dough with 13.9% of *A. diaperinus* powder; L_d1_, dough with 4.6% of *L. migratoria* powder; L_d2_, dough with 9.3% of *L. migratoria* powder; L_d3_, dough with 13.9% of *L. migratoria* powder.

**Figure 4 foods-10-01806-f004:**
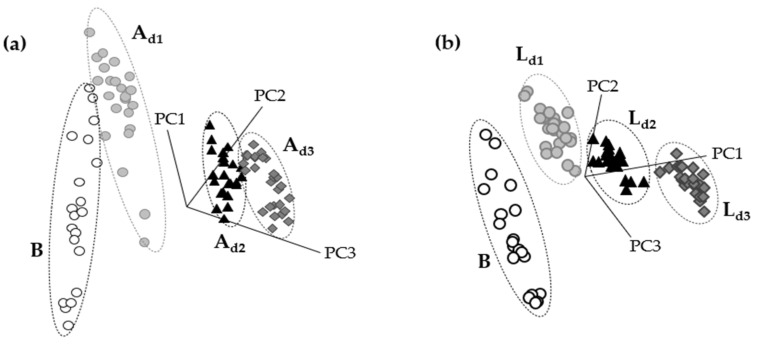
SIMCA class projection plots of (**a**) 4-class SIMCA *A. diaperinus* dough model and (**b**) 4-class SIMCA *L. migratoria* dough model. Abbreviations used: B, blank dough with 0% insect powder; A_d1_, dough with 4.6% of *A. diaperinus* powder; A_d2_, dough with 9.3% of *A. diaperinus* powder; A_d3_, dough with 13.9% of *A. diaperinus* powder.

**Figure 5 foods-10-01806-f005:**
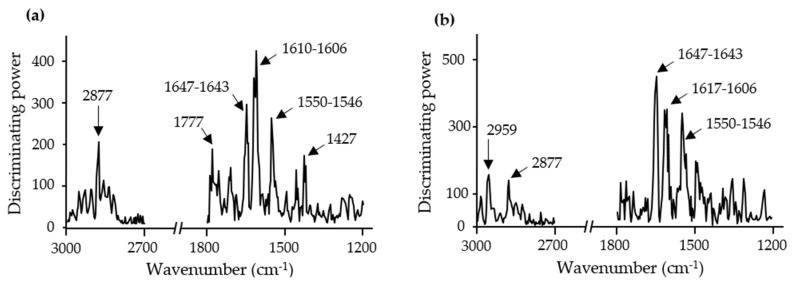
Soft independent modeling of class analogy discriminating power plots of FTIR spectroscopy spectra (3000–2700 cm^−1^ and 1800–1200 cm^−1^) from (**a**) 4-class SIMCA *A. diaperinus* dough model and (**b**) 4-class SIMCA *L. migratoria* dough model.

**Figure 6 foods-10-01806-f006:**
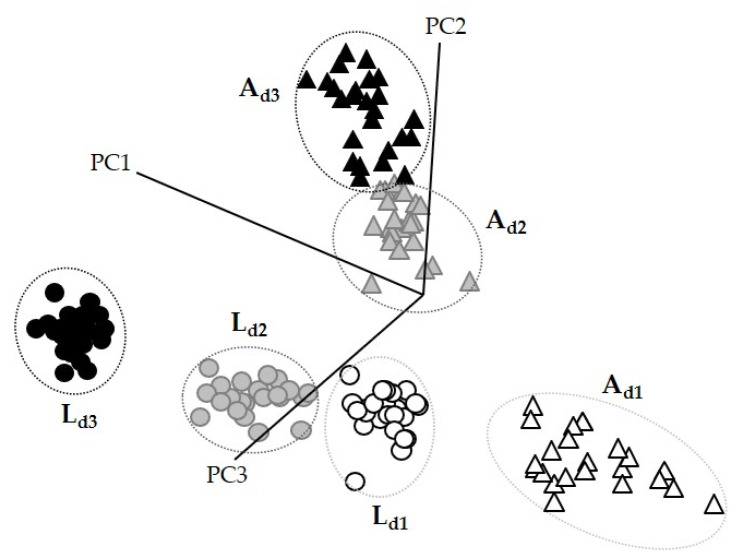
Soft independent modeling of class analogy class projection of FTIR spectroscopy spectra (3000–2700 cm^−1^ and 1800–1200 cm^−1^) from a 6-class SIMCA *A. diaperinus* vs. *L. migratoria* dough model. Abbreviations used: A_d1_, dough with 4.6% of *A. diaperinus* powder; A_d2_, dough with 9.3% of *A. diaperinus* powder; A_d3_, dough with 13.9% of *A. diaperinus* powder; L_d1_, dough with 4.6% of *L. migratoria;* L_d2_, dough with 9.3% of *L. migratoria;* L_d3_, dough with 13.9% of *L. migratoria*.

**Figure 7 foods-10-01806-f007:**
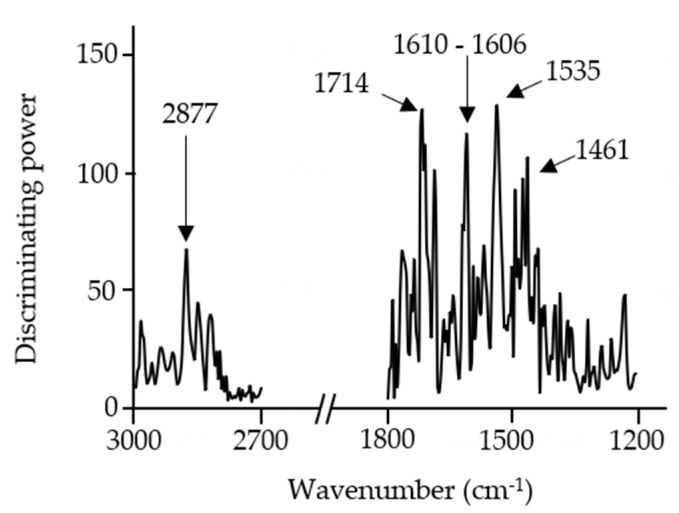
Soft independent modeling of class analogy discriminating power plot of FTIR spectroscopy spectra (3000–2700 cm^−1^ and 1800–700 cm^−1^) from a 6-class SIMCA *A. diaperinus* vs. *L. migratoria* dough model.

**Figure 8 foods-10-01806-f008:**
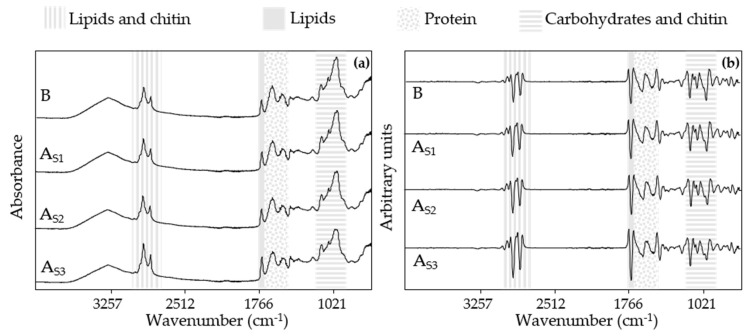
Raw mid-infrared spectra (4000 to 800 cm^−1^) from the snacks made with *A. diaperinus* powder (**a**) and the transformed spectra (MSA and second derivative, 13 points) (**b**). Abbreviations used: Bs, blank snack with 0% insect powder; As_1_, snack with 7.2% of *A. diaperinus* powder; As_2_, snack with 14.4% of *A. diaperinus* powder; As_3_, snack with 21.9% of *A. diaperinus* powder.

**Figure 9 foods-10-01806-f009:**
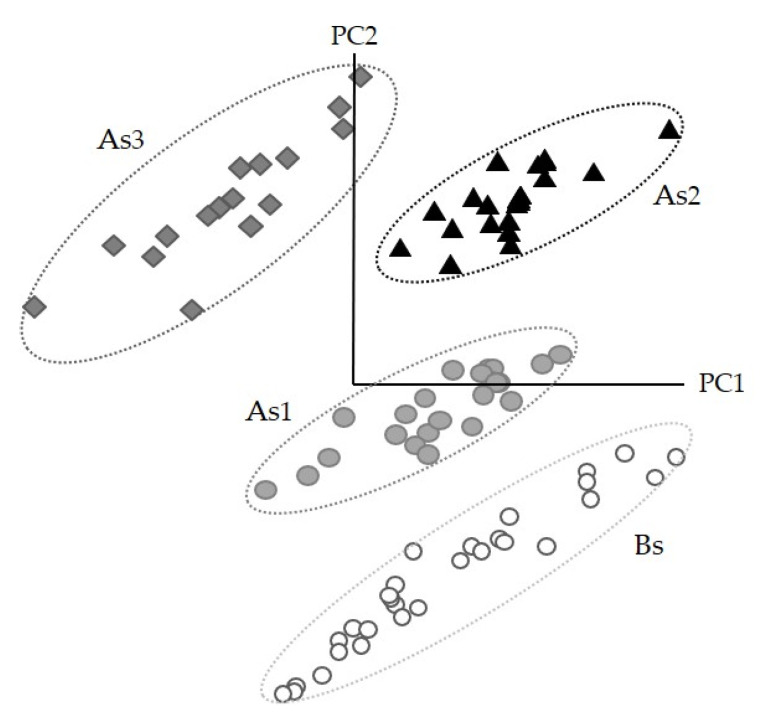
Soft independent modeling of class analogy class projection of FTIR spectroscopy spectra (3000–2700 cm^−1^ and 1800–1200 cm^−1^) from a 4-class SIMCA *A. diaperinus* snack model. Abbreviations used: Bs, blank snack with 0% insect powder; As_1_, snack with 7.2% of *A. diaperinus* powder; As_2_, snack with 14.4% of *A. diaperinus* powder; As_3_, snack with 21.9% of *A. diaperinus* powder.

**Figure 10 foods-10-01806-f010:**
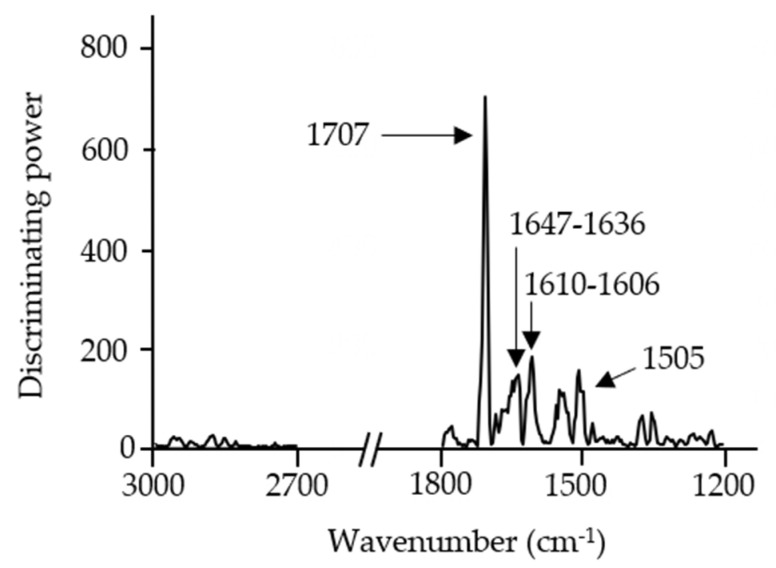
Soft independent modeling of class analogy discriminating power plot of FTIR spectroscopy spectra (3000–2700 cm^−1^ and 1800–1200 cm^−1^) from a 4-class SIMCA *A. diaperinus* snack model.

**Figure 11 foods-10-01806-f011:**
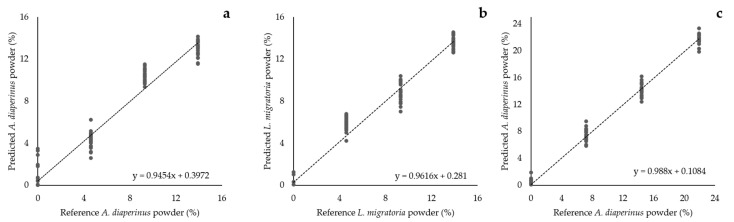
Modeling equations of the calibrated partial least square regressions (PLSR) from the models (**a**) A_d_ PLSR, *A. diaperinus* dough PLSR model; (**b**) L_d_ PLSR, *L. migratoria* dough model; and (**c**) A_s_ PLSR, *A. diaperinus* 3D-printed baked snacks PLSR model.

**Figure 12 foods-10-01806-f012:**
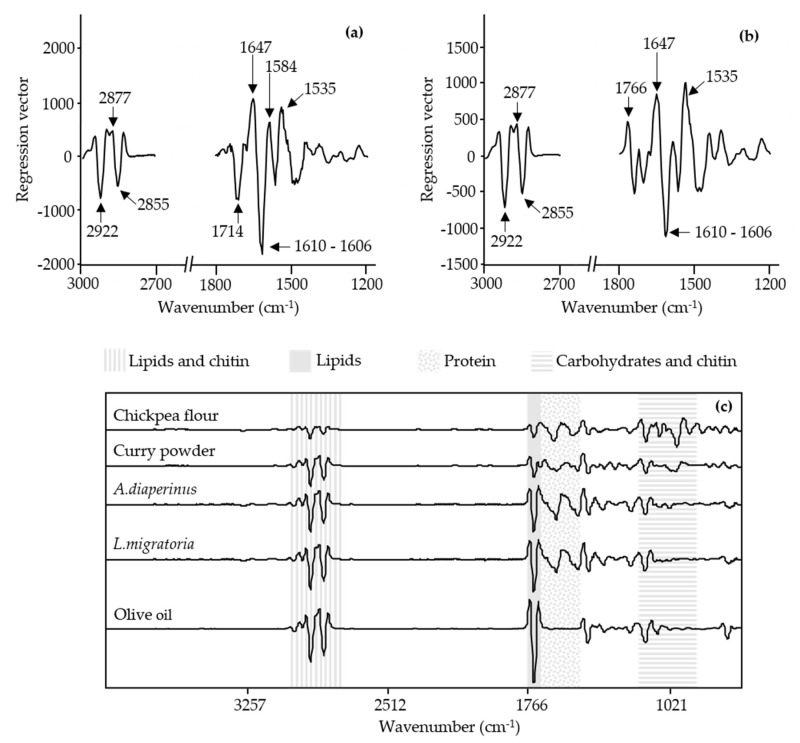
PLSR regression vector plot of FTIR spectroscopy spectra (3000–2700 cm^−1^ and 1800–1200 cm^−1^) from an (**a**) *A. diaperinus* dough prediction model, (**b**) *L. migratoria* dough prediction model, and (**c**) the transformed spectra (vector length normalized and transformed using second derivative polynomial-fit Sa-vitzky–Golay function (13 points)) of the raw ingredients used for the dough preparation.

**Figure 13 foods-10-01806-f013:**
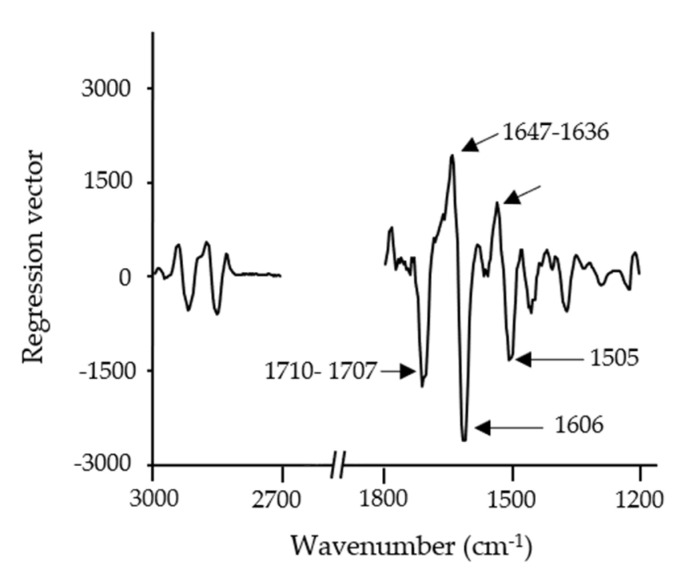
PLSR regression vector plot of FTIR spectroscopy spectra (3000–2700 cm^−1^ and 1800–1200 cm^−1^) from an *A. diaperinus* snack model.

**Table 1 foods-10-01806-t001:** Insect powder percentage and measured weight of each ingredient for different formulations.

Ingredients (g)	Dough Formulation ^a^						
	B	A_d1_	A_d2_	A_d3_	L_d1_	L_d2_	L_d3_
Chickpea flour	100.03 ± 0.01	90.05 ± 0.02	80.02 ± 0.01	70.03 ± 0.01	90.03 ± 0.05	80.03 ± 0.02	70.04 ± 0.05
*A. diaperinus* powder	-	10.02 ± 0.05	20.03 ± 0.04	30.02 ± 0.06	-	-	-
*L. migratoria* powder	-	-	-	-	10.07 ± 0.08	20.08 ± 0.09	30.06 ± 0.08
Water	85.01 ± 0.01	85.01 ± 0.01	85.00 ± 0.01	85.00 ± 0.01	85.01 ± 0.01	85.01 ± 0.01	85.01 ± 0.01
Olive oil	25.00 ± 0.01	25.01 ± 0.01	25.00 ± 0.01	25.01 ± 0.01	25.01 ± 0.01	25.00 ± 0.01	25.00 ± 0.01
Curry powder	4.00 ± 0.01	4.01 ± 0.01	4.00 ± 0.01	4.00 ± 0.01	4.01 ± 0.01	4.00 ± 0.01	4.00 ± 0.01
Salt	2.05 ± 0.03	2.04 ± 0.03	2.06 ± 0.02	2.02 ± 0.05	2.05 ± 0.04	2.07 ± 0.02	2.03 ± 0.03
Insect powder (%)	0.00 ± 0.00	4.64 ± 0.02	9.27 ± 0.02	13.89 ± 0.03	4.65 ± 0.02	9.28 ± 0.02	13.90 ± 0.02

Abbreviations used: B, blank dough with 0% insect powder; A_d1_, dough with 4.6% of *A. diaperinus* powder; A_d2_, dough with 9.3% of *A. diaperinus* powder; A_d3_, dough with 13.9% of *A. diaperinus* powder; L_d1_, dough with 4.6% of *L. migratoria* powder; L_d2_, dough with 9.3% of *L. migratoria* powder; L_d3_, dough with 13.9% of *L. migratoria* powder. ^a^ Values are means of three different batches  ±  standard deviation (SD).

**Table 2 foods-10-01806-t002:** Water loss of the different 3D-printed baked snacks and the concentration of insect powder after the baking process. Abbreviations used: B, blank snack with 0% insect powder; As_1_, snack with 7.2% of *A. diaperinus* powder; As_2_, snack with 14.4% of *A. diaperinus* powder; As_3_, snack with 21.9% of *A. diaperinus* powder.

Snack Formulation	B	As_1_	As_2_	As_3_
Water loss (%) ^a^	36.2 ± 0.2	36.0 ± 0.5	35.5 ± 1.5	36.5 ± 1.2
*A. diaperinus* powder (wt%) ^a^	0.0 ± 0.0	7.2 ± 0.1	14.4 ± 0.4	21.9 ± 0.4

^a^ Values are means of three different batches  ±  standard deviation.

**Table 3 foods-10-01806-t003:** The number of factors, their cumulative variance, and the number of outliers used to create 4-class SIMCA models of *A. diaperinus* and *L. migratoria* dough.

Model	Class	Factor 1 (%)	Factor 2 (%)	Factor 3 (%)	Outliers
4-class SIMCA *A. diaperinus* dough model	B	82.2	92.7	95.0	0
	A_d1_	81.2	92.5	96.3	5
	A_d2_	92.0	95.8	98.0	3
	A_d3_	91.7	97.0	98.5	5
4-class SIMCA *L. migratoria* dough model	B	82.2	95.7	95.0	0
	L_d1_	86.0	91.7	95.7	4
	L_d2_	96.9	98.2	99.1	6
	L_d3_	75.0	85.7	91.9	5

Abbreviations used: B, blank dough with 0% insect powder; A_d1_, dough with 4.6% of *A. diaperinus* powder; A_d2_, dough with 9.3% of *A. diaperinus* powder; A_d3_, dough with 13.9% of *A. diaperinus* powder; L_d1_, dough with 4.6% of *L. migratoria;* L_d2_, dough with 9.3% of *L. migratoria;* L_d3_, dough with 13.9% of *L. migratoria.*

**Table 4 foods-10-01806-t004:** Soft independent modeling of class analogy interclasses distances from the 4-class SIMCA *A. diaperinus* doughs model and the 4-class SIMCA *L. migratoria* doughs model.

***A. diaperinus***	**B**	**A_d1_**	**A_d2_**	**A_d3_**
B	0.0			
A_d1_	3.1	0.0		
A_d2_	8.7	4.5	0.0	
A_d3_	12.2	7.2	3.1	0.0
***L. migratoria***	**B**	**L_d1_**	**L_d2_**	**L_d3_**
B	0.0			
L_d1_	5.8	0.0		
L_d2_	8.4	3.0	0.0	
L_d3_	13.6	7.4	3.9	0.0

Abbreviations used: B, blank dough with 0% insect powder; A_d1_, dough with 4.6% of *A. diaperinus* powder; A_d2_, dough with 9.3% of *A. diaperinus* powder; A_d3_, dough with 13.9% of *A. diaperinus* powder; L_d1_, dough with 4.6% of *L. migratoria;* L_d2_, dough with 9.3% of *L. migratoria;* L_d3_, dough with 13.9% of *L. migratoria.*

**Table 5 foods-10-01806-t005:** Dough model predictions of *A. diaperinus* SIMCA model using *L. migratoria* spectra and *L. migratoria* SIMCA model using *A. diaperinus* spectra.

Model	Dough	No. Spectra	Best Class	Next Best	Not Classified
4-class SIMCA *A. diaperinus*	L_d1_	26	0%	0%	100%
	L_d2_	24	0%	0%	100%
	L_d3_	25	0%	0%	100%
4-class SIMCA *L. migratoria*	A_d1_	25	0%	0%	100%
	A_d2_	27	0%	0%	100%
	A_d3_	25	0%	0%	100%

Abbreviations used: B, blank dough with 0% insect powder; A_d1_, dough with 4.6% of *A. diaperinus* powder; A_d2_, dough with 9.3% of *A. diaperinus* powder; A_d3_, dough with 13.9% of *A. diaperinus* powder; L_d1_, dough with 4.6% of *L. migratoria;* L_d2_, dough with 9.3% of *L. migratoria;* L_d3_, dough with 13.9% of *L. migratoria.*

**Table 6 foods-10-01806-t006:** The number of factors, their cumulative variance, and the number of outliers used to create a 6-class SIMCA model of *A. diaperinus* vs. *L. migratoria* dough.

Model	Class	Factor 1 (%)	Factor 2 (%)	Factor 3 (%)	Outliers
6-class SIMCA *A. diaperinus* vs. *L. migratoria* dough model	L_d1_	86.0	91.7	95.7	4
	A_d1_	83.9	94.7	96.6	6
	L_d2_	96.7	98.2	99.1	7
	A_d2_	92.0	95.8	98.0	6
	L_d3_	75.9	85.7	91.9	5
	A_d3_	91.7	96.9	98.5	5

**Table 7 foods-10-01806-t007:** Soft independent modeling of class analogy (SIMCA) interclasses distances from a 6-class SIMCA *A. diaperinus* vs. *L. migratoria* doughs model.

Dough	L_d1_	A_d1_	L_d2_	A_d2_	L_d3_	A_d3_
L_d1_	0.0					
A_d1_	3.0	0.0				
L_d2_	3.0	4.7	0.0			
A_d2_	3.3	4.3	3.8	0.0		
L_d3_	7.4	9.3	3.9	6.7	0.0	
A_d3_	6.4	7.3	5.1	3.1	5.0	0.0

Abbreviations used: A_d1_, dough with 4.6% of *A. diaperinus* powder; A_d2_, dough with 9.3% of *A. diaperinus* powder; A_d3_, dough with 13.9% of *A. diaperinus* powder; L_d1_, dough with 4.6% of *L. migratoria;* L_d2_, dough with 9.3% of *L. migratoria;* L_d3_, dough with 13.9% of *L. migratoria.*

**Table 8 foods-10-01806-t008:** The number of factors, their cumulative variance, and the number of outliers used to create a 4-class SIMCA model of *A. diaperinus* snacks.

Model	Class	Factor 1 (%)	Factor 2 (%)	Outliers
4-class SIMCA *A. diaperinus* snack model	Bs	97.2	98.7	1
	As_1_	97.0	97.9	6
	As_2_	89.5	93.5	4
	As_3_	93.8	96.9	0

Abbreviations used: Bs, blank snack with 0% insect powder; As_1_, snack with 7.2% of *A. diaperinus* powder; As_2_, snack with 14.4% of *A. diaperinus* powder; As_3_, snack with 21.9% of *A. diaperinus* powder.

**Table 9 foods-10-01806-t009:** Soft independent modeling of class analogy interclass distances from a 4-class SIMCA *A. diaperinus* snack model. Abbreviations used: Bs, blank snack with 0% insect powder; As_1_, snack with 7.2% of *A. diaperinus* powder; As_2_, snack with 14.4% of *A. diaperinus* powder; As_3_, snack with 21.9% of *A. diaperinus* powder.

Snack	Bs	A_S1_	A_S2_	A_S3_
B	0.0			
A_S1_	3.3	0.0		
A_S2_	5.9	3.0	0.0	
A_S3_	8.2	5.1	3.3	0.0

**Table 10 foods-10-01806-t010:** Parameters associated with each partial least square regression (PLSR) model. Abbreviations used: *n* refers to the number of spectra used to create the model; A_d_ PLSR, *A. diaperinus* dough PLSR model; L_d_ PLSR, *L. migratoria* dough model; and A_s_ PLSR, *A. diaperinus* 3D-printed baked snacks PLSR model.

Model	Factors	*n*	Cumulative Variance (%)	SEP	R^2^_val_	SEC	R^2^_cal_
A_d_ PLSR	2	93	96.2	1.24	0.970	1.21	0.972
L_d_ PLSR	2	91	95.6	1.08	0.978	1.06	0.979
A_s_ PLSR	2	89	97.8	0.90	0.994	0.88	0.994

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
