# Peer review of "ATR-FTIR Spectroscopy Combined with Multivariate Analysis Successfully Discriminates Raw Doughs and Baked 3D-Printed Snacks Enriched with Edible Insect Powder"

_foods, 2021, doi:10.3390/foods10081806_

Round 1
Reviewer 1 Report
The manuscript presents the potential of using a portable FTIR device combined with chemometric methods for rapidly discriminating and predicting the concentration of insect powders added into raw dough and 3D printed baked snacks. Based on the presented results, I addressed my comments and suggestions:
1. Section 2.2. Dough Preparation: It is known that blending time affects the grade of heterogeneity of the blends. Do you evaluate other times of blending?
2. In Table 1, replacing using insect powders is curious. The dough formulations were replaced by just one insect powder but not by a combination of them. Why did you decide to use just one for replacement and not a combination?. For example, 5% A and 5% L to a total of 10% (A+L). Following this formulation, what properties do you selected to compare formulation B against the new formulations?
3. Section 2.3. D printing and Post-Processing Analysis: Can you present the results of the average particle size of the formulations? How much was affected the particle size after milling within the electric coffee grinder?
4. Section 2.4. Spectral Data Acquisition by FTIR: How much was prepared for each formulation blend in Table 1? What percentage of the total mass of the dough represented the 4 mg used for spectral acquisition? What procedure did you do to extract this 4 mg? Depending on the heterogeneity of the blends and the extraction for analysis, the quality of the results will be affected significantly. Same question in line 159.
5. Line 180: cross-validation?
6. Line 184: LOO is by far the worse method of cross-validation. However, if you have a prediction set of samples and use the prediction values for model criteria, it is ok to use LOO to create the model as an initial step.
7. Line 191: It is confusing the way of explaining this step. When you say internal validation, do you mean cross-validation, or do you mean a prediction set of samples? When you say new prediction model, specifically, what do you mean? I recommend using the correct terminology to avoid confusion with new readers. I recommend you to read the papers of Kim Esbesen about validation in chemometrics. This is his Scholar profile: https://scholar.google.com/citations?hl=en&user=pJX-qg0AAAAJ
8. Line 192: The reference data is the y-variable, x-variable is the matrix of absorbances for each frequency/wavenumber/wavelength.
9. Line 194: Predicted y-variable.
10. Line 245: This is critical for model performance. If you didn't add variations on the other constituents, it would cause that future samples to fall as outliers with minor variations on that constituents because the model didn't include these variations.
11. Section 4. Conclusions: Based on the design of the experiment (Table 1) and the evaluation performed on the PLS results. The conclusion can not be stated as it is. It is necessary several evaluations to make this statement. Furthermore, most of the manuscript is dedicated to SIMCA models and a brief and scarce section to PLS results.
12. The DoE of the formulations needs to be uncorrelated. The authors changed the amount of two ingredients, chickpea flour, and one insect powder. It is necessary to make changes in all the ingredients breaking the correlation between them.
13. The regression vectors in Figures 12 and 13 need to be compared with the spectra of the constituents in Figure 2 (using the same spectral preprocessing of the PLS models). This comparison will help evaluate if the model is specific to insect powder or probably chickpea flour due these ingredients were changed inversely in their amounts (Table 1).
14. Based on the similarities of both insect powders, the authors need to evaluate the model's specificity for one of the insect powders. It can be done, for example, using the A. diaperinus model and predict the L. migratoria formulation.
15. I recommend reducing the SIMCA section and perform a detailed analysis of PLS results.
Reviewer 2 Report
The manuscript describes a spectroscopic-based solution for discriminating snacks enriched with edible insect powder. In order to achieve this goal, SIMCA was used for classification, and then, PLS has been exploited for quantification.
In general, the manuscript is clear and well-cenceived. My main suggestion for the authors is to increase the number of samples, and use external validation of the models.
Reviewer 3 Report
Manuscript Foods-1301030 deals with the hypothesis that ATR-FT-MIR combined with multivariate analysis can be used as a rapid technique to discriminate edible insect powders used as ingredients in complex formulations such as foodstuffs. The article is very well-written, clear and results are given in details. The methodology used to achieve the purpose of the study and statistical analysis is properly given and discussed. Predicting models have also been developed. The study is an important contribution to the field. Minor corrections are needed. These are;
-Line 400. ‘’As it has been commented before….’’
-Line 407. Kindly correct the citation format.
-Graphical abstract
Graphical abstract is very useful for the readers.
Based on the above, I suggest a minor revision.
Round 2
Reviewer 1 Report
The authors addressed most of all the comments; just a few questions about their responses. The questions are following the same numbering as the first review:
1. Can you include the properties used and their measurements? It is not necessary a table with all the measurements. You can indicate which values were optimal for your study.
3. I did not see the particle size figures and paragraphs added within the revised version of the manuscript.
13. I agree with the authors with part of their comments. However, the comparison provided by them carries to the readers to look at two figures separated far from one to another. This can be avoided by showing the comparison in one single figure. The comparison of regression vector vs. the spectra of constituents helps to visualize what authors write on the text.
14. I did not see the new table and paragraphs added within the revised version of the manuscript.
15. I do not agree with the authors regarding providing all the important outputs for the PLS section. The authors did not evaluate several figures of merit (the FOM can be read in the Kim Esbensen papers). The development of a PLS model is not straightforward, as it seems in this study. The authors did not break the correlation of the constituents, and this is crucial for model development. Even without considering the effect in blend properties, a low correlation is necessary to avoid overfitting or poor interpretation of the results. However, this study shows the feasibility of doing an in-depth study with the evaluation of not correlated DoE, the effect of formulation properties, the effect of the sampling methodology, and the FOM of the models.
I did not see the modified figures 12 and 13 within the revised version of the manuscript.
Reviewer 2 Report
.
Author Response
We did not have more comments to address from this reviewer from this second round. The document attached did not have any comment.